# Growth Restriction of *Rhizoctonia solani* via Breakage of Intracellular Organelles Using Crude Extracts of Gallnut and Clove

**DOI:** 10.3390/molecules26061667

**Published:** 2021-03-17

**Authors:** Jian Wang, Xianfeng Hu, Chenglong Yang, Xiaomao Wu, Rongyu Li, Ming Li

**Affiliations:** 1Institute of Crop Protection, Guizhou University, Guiyang 550025, China; Wangj_ian@sina.com (J.W.); huxianfenggzu@163.com (X.H.); wuxm827@126.com (X.W.); 2Institute of Subtropical Crops, Guizhou Academy of Agricultural Sciences, Xingyi 562400, China; yangchenglong208@163.com; 3The Provincial Key Laboratory for Agricultural Pest Management in Mountainous Region, Guizhou University, Guiyang 550025, China

**Keywords:** *Rhus chinensis* Mill, *Syzygium aromaticum*, rice sheath blight, antifungal activity

## Abstract

Plant diseases reduce crop yield and quality, hampering the development of agriculture. Fungicides, which restrict chemical synthesis in fungi, are the strongest controls for plant diseases. However, the harmful effects on the environment due to continued and uncontrolled utilization of fungicides have become a major challenge in recent years. Plant-sourced fungicides are a class of plant antibacterial substances or compounds that induce plant defenses. They can kill or inhibit the growth of target pathogens efficiently with no or low toxicity, they degrade readily, and do not prompt development of resistance, which has led to their widespread use. In this study, the growth inhibition effect of 24 plant-sourced ethanol extracts on rice sprigs was studied. Ethanol extract of gallnuts and cloves inhibited the growth of bacteria by up to 100%. Indoor toxicity measurement results showed that the gallnut and glove constituents inhibition reached 39.23 μg/mL and 18.82 μg/mL, respectively. Extract treated rice sprigs were dry and wrinkled. Gallnut caused intracellular swelling and breakage of mitochondria, disintegration of nuclei, aggregation of protoplasts, and complete degradation of organelles in hyphae and aggregation of cellular contents. Protection of *Rhizoctonia solani* viability reached 46.8% for gallnut and 37.88% for clove in water emulsions of 1000 μg/mL gallnut and clove in the presence of 0.1% Tween 80. The protection by gallnut was significantly stronger than that of clove. The data could inform the choice of plant-sourced fungicides for the comprehensive treatment of rice sprig disease. The studied extract effectively protected rice sprigs and could be a suitable alternative to commercially available chemical fungicides. Further optimized field trials are needed to effectively sterilize rice paddies.

## 1. Introduction

Rice sheath blight, known as Moire disease, is one of the most serious fungal diseases or rice crops globally [1,2]. Sheath blight, caused by *Rhizoctonia solani* Kühn AG1 1A (teleomorph *Thanatephorus cucumeris* (A. B. Frank) Donk), is one of the most important diseases in rice worldwide [3,4]. Rice sheath blight affects both the quality and yield of rice. The prevalence if rice sheath blight is increasing in China, reflecting changes in rice farming that include increased planting density and lack of high resistance varieties. In high-temperature and humid environments, rice sheath blight has reduced rice yields by up to 50% [2,3,4,5]. Rice sheath blight hosts are widespread and the fungus core can survive for a long time in the soil, with a high rate of genetic variation. These factors make it difficult to control the disease in the production process [6].

The lack of rice varieties with high resistance to rice sheath blight complicates the prevention and treatment. Current measures are mainly chemical and most commonly include tyrutin and mimide fluoroazoles. Jinggangmycin, which is produced from water-absorbing streptomycin bacteria, is most commonly used in China [7,8]. Resistance was first described in Zhengzhou County, Henan Province [9]. The field statistical resistance rate of Fujian Province reached 2.48% in 2015 [10]. Surveillance of jinggangmycin resistance in 206 rice sheath blight strains in 26 districts of 12 cities found that the strains became resistant, with increasing dosages required year-by-year [11,12]. Additional new problems, including environmental pollution, deterioration of human health, damage to non-target organisms in the field, destruction of the ecological balance of rice paddies, and serious problems caused by the utilization of jinggangmycin, led the European Union to ban the use of jinggangmycin in 2002 (The Commission of European Communities, No. 2076/2002). Accordingly, there is a need for plant-sourced environmentally friendly pesticides with high efficiency, low toxicity, and broad spectrum activity.

Important facets of biosynthetic pesticides attributed to plant-sourced pesticides include high efficiency, low or no toxicity, easy degradation, and lack of development of resistance to drugs. Antibacterial substances in plants or the induction of plant defenses can kill or inhibit the growth of pathogenic bacteria. More than 500,000 plant species have been identified globally. However, only about 10% of known plants’ chemical compositions have been analyzed [13]. Grange et al. reported that 2400 plants have active ingredients for pest control [14]. Plant resources are extraordinarily rich in China, with more than 10,000 kinds of Chinese herbal medicine resources, 1400 kinds of plants with inhibitory activity, and over 328,000 plant-based natural compounds [13,15]. Fogliani studied the antibacterial activity of 50 species of fire-barrel tree plants from Scotland and reported that 49 inhibited spores to varying extents [16]. Ojala et al. showed that extracts of wrinkled parsley and fennel have antibacterial effects on antiseptic sclerosis [17]. Srinivasan et al. studied the antibacterial activity of 50 medicinal plant water products in India and found that 36 of the plants had antibacterial activity and 12 had broad spectrum antibacterial effects [18].

Plant-sourced fungicides are an important part of biorational pesticides and have become the focus of the current research. In this study, the growth inhibition effect of 24 plant-sourced active compounds on the growth of mycelia of rice sheath blight was studied. Indoor toxicology and viability protection studies revealed that gallnut and clove protected rice by inhibiting the mycelial growth of rice sheath blight. Further inhibition mechanisms were investigated by transmission electron microscopy and scanning electron microscopy. These findings demonstrated that internal cell structures of rice sheath blight were destroyed and the internal milieu was disordered. The results should inform new research strategies for the prevention and control of rice sheath blight and are expected to lead to the development of new botanical fungicides.

## 2. Results

### 2.1. Inhibition of R. solani Plant-Derived Extracts

The biological control efficiency of 24 plant ethanol extracts against *R. solani* was analyzed at room temperature (Table 1). The incidence and inhibition percentage of all plant-based ethanol extracts were analyzed. The 24 plant ethanol extracts tested at a concentration of 10 mg/mL showed varying degrees of inhibition of *R. solani*. The inhibition effect of the ethanol extract of gallnut and cloves was the best, with antibacterial rates of 100%. They were followed by the ethanol extracts of Chinese angelica and Cinnamon, with rates of 51.31% and 66.99%, respectively. The remaining ethanol extracts produced rates with a content of <30%. Gallnut and clove extracts were selected for further analyses.

### 2.2. Indoor Toxicity Test of Plant-Derived Extracts against R. solani

Indoor toxicity tests were performed for the ethanol extracts of cloves and gallnut, because both completely inhibited *R. solani* at a concentration of 10 mg/mL. When the concentration of the ethanol extracts ranged from 10 to 75 μg/mL, the inhibition rate of ethanol extracts of clove at 50 μg/mL was 89.85%, and gallnut ethanol extract at 50 μg/mL was only 77.66%. The findings demonstrated that a low concentration of ethanol extract of clove effectively inhibited *R. solani* mycelium. With increasing extract concentration, a significant difference between the two extracts was observed (*p* < 0.05, Table 2, Figure 1).

Statistical analysis of the data revealed a significant linear relationship between the concentration of the extract and the inhibition rate, with correlation coefficients of 0.9678 for gallnut and 0.9593 for clove. The EC_50_ values of the ethanol extracts of gallnut and clove against *R. solani* were 41.84 and 21.68 μg/mL, respectively. The EC_75_ values was 72.59 and 36.34 μg/mL, respectively. The results showed that both gallnut and clove extracts showed good effects (Table 3). 

### 2.3. Effects of Ethanol Extracts from Clove and Gallnut on Mycelial Morphology of R. solani

The colony morphology of *R. solani* treated with different concentrations of ethanol extracts of gallnut and clove changed significantly. Scanning electron microscopy (SEM) revealed that the mycelia of the control treatment were uniform in diameter with a smooth surface, good extension, and complete secondary mycelium growth (Figure 2a,b). In contrast, a 24 h treatment with 20 μg/mL clove extract produced seriously shriveled mycelia with thinner growth point of secondary mycelium (Figure 2c,d). After treatment with 40 μg/mL gallnut extract (Figure 2e,f) for 24 h, the mycelium of *R. solani* was deformed, shriveled, and folded, and the thickness was uneven. The growth point of the secondary mycelium was damaged and folded. Both extracts produced shriveled and shrivel mycelia. The effect of the clove ethanol extract at low concentrations was more pronounced than that of gallnut ethanol extract.

### 2.4. Effects of Ethanol Extracts from Clove and Gallnut on Ultrastructure of Mycelia of Rice Sheath Blight

The mycelia of *R. solani* treated with ethanol extracts of gallnut and clove displayed significant changes in morphology and profound damage of cell ultrastructure. In particular, the permeability of mycelial cells and the structure and morphology of organelles, such as the cytoplasm and mitochondria, were significantly altered, as shown in Figure 3. The cell morphology of the control (Figure 3a,b) was regular and the structure was complete. The cell wall was uniform in texture and thickness, and was closely linked to the cell membrane. The protoplast was dense and uniform. The mitochondria, endoplasmic reticulum, vacuole, and other structures were clear and complete. Mycelia treated with 72.59 μg/mL gallnut ethanol extract displayed swelling and rupture of mitochondria, disintegrated nuclei, and aggregation of protoplasts (Figure 3c,d; Table 3). The organelles in mycelia treated with 36.34 μg/mL ethanol extract of clove were completely degraded with aggregated cell contents (Figure 3e,f; Table 3).

### 2.5. Protective Effects of Clove and Gallnut Ethanol Extracts on Rice In Vivo

The control effect increased as the concentration of the ethanol extracts of gallnut and clove increased. The control effect of gallnut reached 46.80% when the ethanol extract concentration ranged between 200 and 1000 μg/mL. The value for clove was only 37.88% at the same concentration range. The control effect of gallnut ethanol extract was 8.92% higher than that of clove at 1000 μg/mL. Thus, although the clove ethanol extract showed a good inhibitory effect on *R. solani*, it did not show a good control effect on rice plants (Table 4, Figure 4). 

### 2.6. LC–MS and Data Analysis

The quantitative ion chromatogram of lauric acid was obtained by liquid chromatography–mass spectrometry (Figure 5a). The peak time of lauric acid was about 7.584 min, the regression equation was Y = 327,272.248x − 254,692.2965, and the correlation coefficient was 0.9987, which indicated that there was a good linear relationship between the concentration and the corresponding peak area in the concentration range of 10–100 μgmL. The lauric acid content in the ethanol extract of gallnut (10 mg/mL) was 1.39 mg/mL, Therefore, the lauric acid content in ethanol extracts of *Galla chinensis* is about 13.9%.

The quantitative ion chromatogram (Figure 5b) of eugenol was obtained by liquid chromatography–mass spectrometry. The peak time of eugenol was 4.012 min. The regression equation was Y = 67,959x + 40,242 and the correlation coefficient was 0.9988, which indicated that there was a good linear relationship between the concentration and the corresponding peak area in the range of 1–100 μg/mL. The concentration of clove extract was 10 mg/mL; therefore, the detection of eugenol was 4.43 mg/mL and the content of eugenol in clove extract was about 44.3%.

## 3. Discussion

Rice sheath blight is one of the main diseases of rice. This disease serious affects rice yield. There are few highly resistant rice varieties [18]. Thus, chemical control is the main approach to disease control. The use of chemical agents approach is hindered by the “3R” problem. The impact on the environment and human health is a research priority. Among the numerous fungicides used to control rice sheath blight, jinggangmycin and its related improved products occupy an important market position. They have become the agricultural microbial antibiotics with the highest sales [19]. However, negative impacts have spurred international appeals for prohibition. Therefore, the development of new biological pesticides to replace jinggangmycin has become a priority.

Plants are an important source of botanical fungicides and an important part of the development of environmentally harmonious pesticides and biorational pesticides. Advantages of plant-derived fungicides include pronounced efficiency, low toxicity, ease of degradation, and lack of development of resistance [20]. Compounds isolated from plant extracts, such as steroids, tannins, flavonoids, alkaloids, and saponins, have antibacterial activity [21]. Venkateswarlu et al. [22] reported that 2% preparations derived from *Zanthoxylum bungeanum* and *Andrographis paniculata* had good inhibitory effects on the sclerotium of stem rot. Lu et al. [23] studied the antifungal activities of the crude extracts of four Chinese herbal medicines and found that the minimum inhibitory concentrations of gallnut and coca seed were both 3.9 μg/mL, with strong inhibitory effects on *Candida albicans* and *Cryptococcus neoformans*. Serrano et al. [24] treated sweet cherry with thymol, eugenol, and menthol, and found that the number of mold, yeast, and aerobic mesophilic bacteria decreased, especially mold and yeast.

In this study, 24 plant-derived extracts were selected to study the biological activity of *R. solani*. Only a few of the extracts displayed inhibitory activity on *R. solani*. In particular, the ethanol extracts of gallnut and clove at 10 mg/mL showed strong inhibitory effects followed by the ethanol extracts of *A. sinensis* and *C. cassia*. The bacteriostatic rate of 50% increased as the extract concentration increased. The ethanol extracts of gallnut and clove displayed good antibacterial effects (EC_50_ of 41.84 and 21.68 μg/mL, respectively).

The gallnut and clove extracts displayed inhibitory effects on the mycelia of rice sheath blight. SEM showed that both extracts produced wrinkles on the surface of the mycelium and caused varying degrees of damage to the secondary growth point of the mycelium. TEM was used to observe the internal structures of the mycelium cells. After treatment with gallnut extract, the mitochondria in the mycelium were swollen and broken, the nuclei of cells were disintegrated, and protoplasts were aggregated. Gallnut reportedly had a strong inhibitory effect on the β-galactosidase activity of *Agrobacterium tumefaciens* strain A136, with an inhibition rate of 33.20% [25]. Gallnut extract also inhibited enamel demineralization in vitro [26], and the survival of *Vibrio parahaemolyticus* and *Listeria monocytogenes* in cooked shrimp and raw tuna [27]. The most prevalent compound on gallnut is tannic acid [28,29], but its inhibitory effect on rice sheath blight hyphae is not significant. Lauric acid, another component of gallnut, is the most active compound for inhibiting *R. solani* hyphae growth (this part of the data will be published at a later date). Li et al. [27,30] found that citral extracted from *Litsea cubeba* could destroy the integrity of the cell wall and membrane permeability of rice blast, resulting in physiological changes and cytotoxicity. Presently, after treatment with clove extract, organelles in the mycelia of *R. solani* were completely degraded, cell contents were aggregated, and the cell wall was ablated. Eugenol is one of the main components of clove. The compound has a minimal inhibitory concentration of 200 μg/mL against mycelia of *Phytophthora nicotianae*. Eugenol can significantly destroy the cell membrane of mycelium without affecting the integrity of the spore membrane [31]. Pasqua et al. [32] and Helander et al. [33] proposed that eugenol could inhibit the production of essential enzymes in bacteria and cause cell wall damage. Zambonelli et al. [34] reported that the cell morphology of fungal hyphae of *R. solani* and *Colletotrichum gloeosporioides* was characterized by increased cytoplasmic vacuoles, accumulation of liposomes, plasma membrane fluctuations, and mitochondrial and endoplasmic reticulum changes. Eugenol is also a volatile essential oil. It can ablate intracellular organelles of *R. solani*, similar to the mode of action of thymol on fungal cells. The site of action of eugenol on mycelial cells is the cell membrane [35,36], yet the ultrastructural damage of the mycelia of rice sheath blight mainly involves intracellular organelles.

When the concentrations of gallnut and clove ethanol extracts were 1000 μg/mL, the control effects were 45.8% and 37.88%, respectively. The effect of the gallnut extract was better than that of the clove extract. This was likely because the clove ethanol extract contains the volatile component clove phenol C_10_H_12_O_2_ (2-methoxy-4-allylphenol). It adopts an allyl chain structure to replace o-methoxyphenol. It is chemically unstable; thus improved stability will be necessary for its optimal use in control of plant diseases [35,36,37]. Gallnut extract displayed a good protective effect against rice sheath blight in vivo. Investigations of the effect of eugenol on *P. nicotianae* in vivo revealed an effective reduction of the incidence of *P. nicotianae* and a good control effect in the field [31]. *Acorus gramineus* extract had good antibacterial activity, and the control effect on rice sheath blight was 25% at 500 μg/mL [38]. Khoa et al. [39] reported that seed soaking and spraying extracts from dry or fresh leaves of *Chromolaena odorata* can reduce the occurrence of sheath blight. The collective observations conclusively indicate that plant extracts can effectively control crop diseases.

## 4. Materials and Methods

### 4.1. Materials

#### 4.1.1. Isolation and Identification of Rice Sheath Blight Pathogen

The *Rhizoctonia solani* AG1 IA strain of rice sheath blight was isolated from infected rice. The strain was identified by the laboratory of the Crop Protection Institute of Guizhou University. *R. solani* was inoculated into potato dextrose agar (PDA) (Shanghai Aladdin Biochemical Technology Co., Ltd., Shanghai, China) and incubated at 25 °C and 150 rpm for 3 days, and stored at −80 °C.

#### 4.1.2. Collection of Plant-Derived Materials

Raw materials of 24 species were purchased in September 2018 from YiPing Company, Huaxi District, Guiyang City, Guizhou, China. Details are provided in Table 1.

### 4.2. Methods

#### 4.2.1. Collection and Preparation of Plant-Derived Extracts

The crude plant extract was obtained using the organic solvent ethanol immersion method modified from a previous study [16]. The air-dried plant was ground into a powder with a shredder. One hundred grams of the powder was added to 250 mL anhydrous ethanol (Shanghai Aladdin Biochemical Technology Co., Ltd., Shanghai, China) at 50 °C for 2 h ultrasonic extraction. The extraction was repeated three times. The combined clear supernatant was filtered using a Buchner funnel and then steamed with a rotary evaporator at 46 °C in water bath solvent. The crude extract was stored in a refrigerator at 4 °C.

#### 4.2.2. In Vitro Evaluation of Rice Sheath Blight Inhibition Activity of Plant-Derived Extracts

The inhibitory effect of the plant-derived ethanol extract on the growth of rice sprigs was determined by the growth rate of mycelia. The herbal extract for testing was first weighed and then completely dissolved in 200 μL of aqueous ethanol. The final concentration of the solution was adjusted to 10 mg/mL with sterile water. The solution was added to PDA medium, which was cooled to approximately 50 °C in a 1:9 volume ratio. The volume was poured into a petri dish and solidified. The procedure was repeated three times, with the same proportion of sterile water and PDA. A 5 mm punch was used to create blocks from plates cultured with the *R. solani* AG1 IA. The punched blocks were carefully added to the center of the drug-containing culture using an inoculation needle, with the mycelium facing down, with one piece added per dish. The samples were cultured for 3 days at 25 °C in a constant-temperature Foster box. The diameter of the bacteria was measured by the cross-cutting method. The average value was determined and the antibacterial rate was calculated as follows:(1)$$\mathrm{Inhibition}\;\mathrm{rate}\;(\%)=\frac{Mc-Mt}{Mc-0.5}\times 100$$
where *Mc* and *Mt* represent the mycelial growth diameter in control, gallnuts-treated, and cloves-treated conditions, respectively. The EC_50_ (effective dose for 50% inhibition) values were estimated statistically by probit analysis with the probit package of SPSS 22.0 software (SPSS Inc., Chicago, IL, USA) [40].

#### 4.2.3. Determination of Eugenol and Lauric Acid Content

Determination of eugenol content: Analysis of eugenol content was performed on a Waters UPLC BEHC18. Water with 0.1% (*v*/*v*) ammonia (A) (Wuhan Chuangsheng Chemicals Co. Ltd., Wuhan, China) and acetonitrile (B) (Wuhan Chuangsheng Chemicals Co. Ltd., Wuhan, China) were used as solvents at a flow rate of 0.5 mL/min. The substances were eluted with a linear gradient as follows: 0.5 min, 25% B; 0.5–2.5 min, 5% B; 2.5–3 min, 5% B, 3–4 min, 5% B; 4–5 min, 95% B; 5–6 min, 95% B; 6–7min, 25% B; 7–8min, 25% B. The ion source was an electrospray ion source in negative ion mode (ESI^-^). Multi-reaction monitoring mode (MRM), capillary 1.48 kV, cone voltage 29 V, collision gas flow rate 0.15 mL/min, the multi-reaction monitoring ion pair and conditions are shown in Table 5. The standard curve of eugenol was drawn according to the mass concentration. The extracted eugenol samples were treated according to the detection method, and the eugenol content was calculated according to the standard curve.

Determination of lauric acid content: A Waters UPLC BEHC18 column was used, with an injection volume of 10 µL and a flow rate of 0.5 mL/min. The mobile phase was 0.1% formic acid water, the mobile phase B was methanol, gradient elution was used, and the gradient elution procedure was as follows: 1 min, 10% B; 1–3min, 10%B; 3–12min, 90% B, 12–13min, 10% B. The ion source was the electrospray ion source in negative ion mode (ESI^-^). Multi-reaction monitoring mode (MRM), capillary 1.48 kV, cone voltage 75 V, collision gas flow rate 0.15 mL/min, the multi-reaction monitoring ion pair and conditions are shown in Table 5. The standard curve of lauric acid was drawn according to the mass concentration. The extracted eugenol samples were treated according to the detection method, and the lauric acid content was calculated according to the standard curve.

#### 4.2.4. Indoor Toxicity Determination of Plant-Derived Extracts

Based on the preceding results, clove and gallnut (Wuhan Chuangsheng Chemicals Co. Ltd., Wuhan, China), which displayed 100% antibacterial rates, were selected for further toxicity tests. Clove extracts were 10, 20, 30, 40, and 50 μg/mL. Gallnut extracts were set as 15, 30, 45, 60, and 75 μg/mL. The detailed antibacterial effects were also analyzed.

#### 4.2.5. Scanning Electron Microscopy (SEM)

*R. solani* AG1 IA was cultured on medium containing 0 or 36.34 μg/mL (EC75 value) of clove ethanol extract and 72.59 μg/mL (EC75 value) gallnut ethanol extract for 3 days. Rectangular blocks (0.5 cm × 0.3 cm) from the edge of the mycelium were placed in a centrifuge tube with 1 mL of 25% dialdehyde fixation fluid. Three blocks were taken for each treatment. Each sample was suctioned repeatedly with a 50 mL syringe until the bubbles on the surface of the mycelium disappeared. The centrifuge tube was sealed and stored overnight at 4 °C. After suction, the retaining fluid was carefully rinsed three times with 0.1 M Phosphate Buffer Solution (PBS) for 10 min each time. Then, 0.5 mL of 1% nitric acid fixative was added within 2 h. Each sample was washed three times with PBS. Ethanol solutions of 30%, 50%, 70%, 80%, and 90% were used for dehydration for 10 min each time, followed by dehydration twice with waterless ethanol for 10 min each time. After dehydration, the specimens were dried in a freeze drier (LGJ-10D; Beijing Fourth Ring Scientific Instrument Co., Ltd., Beijing, China), and sputter-coated with gold. Microscopy was performed using an SEM (S-3400N; Hitachi, Tokyo, Japan) operated at an accelerating voltage of 20 kV. Controls consisted of untreated mycelia, which were prepared in parallel with experimental samples.

#### 4.2.6. Transmission Electron Microscopy (TEM)

The mycelium collected as described above was poured into a centrifuge tube with 1 mL of 2.5% dialdehyde fixation fluid. The tube was sealed and incubated overnight at 4 °C. After the remaining fluid was carefully suctioned off, the sample was rinsed three times with 0.1 M PBS, for 10 min each time. Then, 0.5 mL of 1% nitric acid fixative was added within 2 h. They were then washed three times with PBS, and ethanol solutions with concentrations of 30%, 50%, 70%, 80%, and 90% were dehydrated for 10 min each time, and then dehydrated twice with waterless ethanol for 20 min each time. A mixture of acetone and resin with different concentration ratios (3:1, 1:1, 1:3 *v*/*v*) was successively used for infiltration for 3 h each time. This was followed by treatment with pure resin overnight. After polymerization at 70 °C for 24 h, the embedded samples were removed for preparation of ultra-thin sections. The sections were stained with lead citrate and uranium diacetate, dried, and observed by TEM (JEM-F200; JEOL, Tokyo, Japan).

#### 4.2.7. Protective Effect of Plant-Derived Extracts on Rice In Vivo

Rice was sown in a plastic basin with a diameter of 16 cm and a height of 12 cm. At the tillering stage, the ethanol extracts of gallnut and clove were diluted with a 0.1% Tween 80 emulsified water solution to 200, 400, 600, 800, or 1000 μg/mL. Each preparation was sprayed on rice plants with 0.1% Tween 80 emulsified water solution as the blank control. After 24 h of spraying, the rice sprigs were inoculated with *R. solani* AG1 IA embedded in a cake (0.6 cm in diameter) on the leaves of the penultimate leaf of the plant. Twenty plants were inoculated per treatment and were cultured in an incubator at a temperature of 25 ± 2 °C, relative humidity of 90%, and 12 h of lighting time. After 7 days of inoculation, the data were acquired according to the classification standard of rice sheath blight, the average value was determined, and the control effect was calculated.

#### 4.2.8. Data Processing and Analysis

The growth inhibition rate was converted to an inhibition probability value. The toxicity regression equation, inhibition medium concentration (EC_50_), and correlation coefficient (R) were calculated using the logarithm of extract concentration and inhibition probability value. Excel software and SPSS statistical software package release 22.0 (SPSS Inc., Chicago, IL, USA) were used to process the relevant data.

## 5. Conclusions

Plant-sourced fungicides are a class of plant antibacterial substances or compounds that induce plant defenses. They can kill or inhibit the growth of target pathogens efficiently with no or low toxicity, degrade readily, and do not prompt development of resistance, which has led to their widespread use. In this study, the growth inhibition effect of 24 plant-sourced ethanol extracts on hyphae of rice sheath blight was studied. Ethanol extracts of gallnuts and cloves inhibited the growth of bacteria by up to 100%. Indoor toxicity measurement results showed that the effective concentrations of gallnut and clove constituents reached 39.23 μg/mL and 18.82 μg/mL, respectively. The results showed that protection from *Rhizoctonia solani* reached 46.8% for gallnut and 37.88% for clove in water emulsions of 1000 μg/mL gallnut and clove in the presence of 0.1% Tween 80. The protective effects of gallnut were significantly stronger than those of clove on rice sheath blight. Although these extracts inhibit the mycelium of rice sheath blight and improve plant defenses, they have not yet been developed as suitable substitutes for commercial chemical fungicides for application in rice agriculture. To achieve this, optimized field trials are needed to determine the optimal application doses in paddy fields.

## Figures and Tables

**Figure 1 molecules-26-01667-f001:**
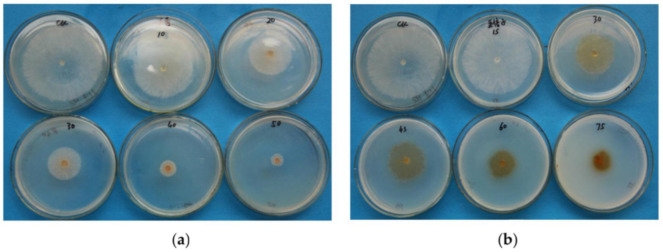
(**a**) The ethanol extracts from clove at concentrations from 10 to 50 µg/mL; (**b**) the ethanol extracts from gallnut at concentrations from 15 to 75 µg/mL.

**Figure 2 molecules-26-01667-f002:**
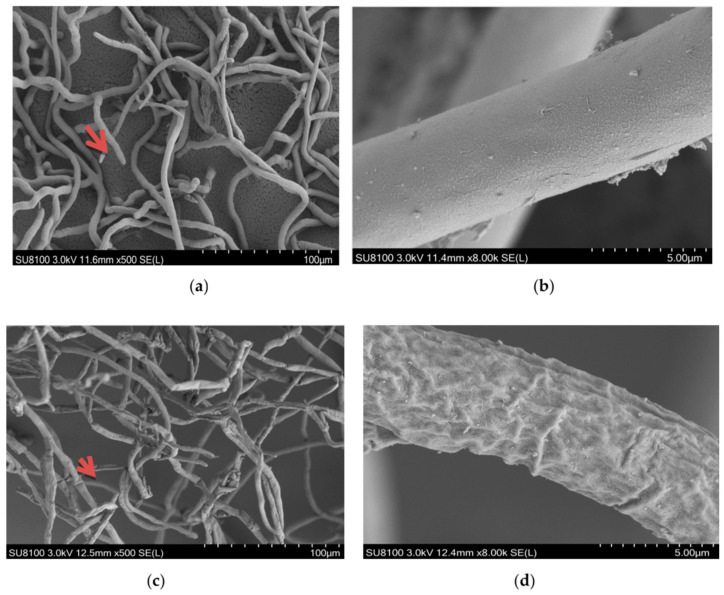
Scanning electron micrographs (SEM) of *R. solani*: hyphae exposed to the ethanol extracts from two *R. solani* at concentrations of (**a**,**b**) 0 µg/mL and (**c**,**d**) EC_75_ = 36.34 µg/mL; the ethanol extracts from clove, (**e**,**f**) EC_75_ = 72.59 µg/mL; the ethanol extracts from gallnut. Arrows and arrowheads indicate hyphae shrinkage and partial distortion.

**Figure 3 molecules-26-01667-f003:**
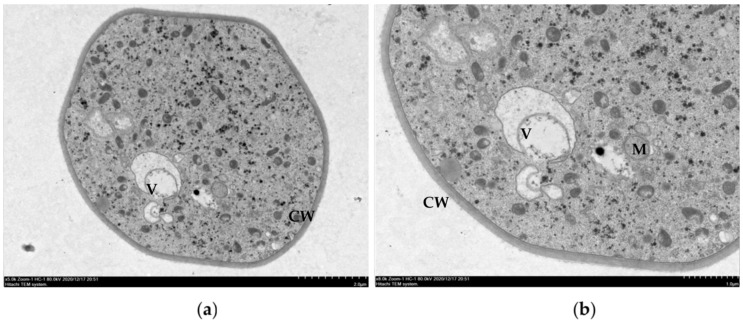
Transmission electron micrographs of *R. solani* hyphae, where hyphae was exposed to the ethanol extracts from two plant at concentrations of (**a**,**b**) 0 µg/mL; (**c**,**d**) EC_75_ = 72.59 µg/m, the ethanol extracts from gallnut; and (**e**,**f**) EC_75_ = 36.34 µg/mL, the ethanol extracts from clove, L; CW = cell wall.

**Figure 4 molecules-26-01667-f004:**
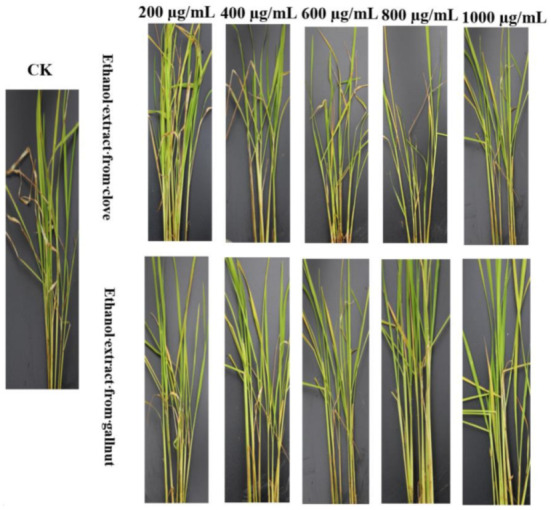
Effect of ethanol extracts from two Chinese herbs on the rice sheath blight. CK: Not treated with drug; The ethanol extracts of gallnut and clove were of different concentrations: 200 μg/mL, 400 μg/mL, 600 μg/mL, 800 μg/mL, 800 μg/mL, 1000 μg/mL.

**Figure 5 molecules-26-01667-f005:**
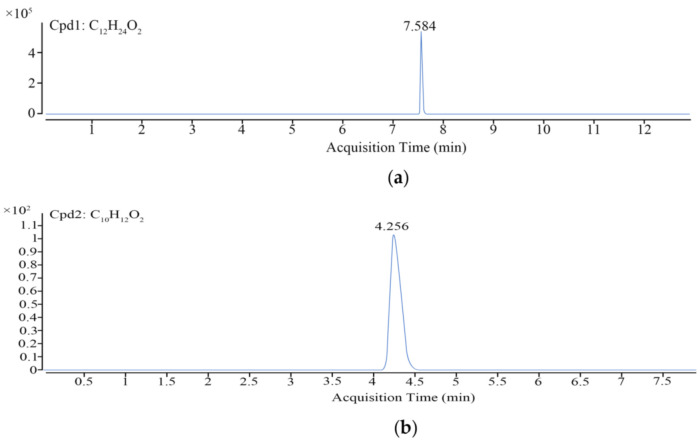
LC–MS analysis. (**a**) Quantitative ion chromatogram of lauric acid; (**b**) Quantitative ion chromatogram of eugenol.

**Table 1 molecules-26-01667-t001:** Inhibition effects of ethanol extracts from plants against *R.solani*.

Plants Resources	Families	Extract Position	Colony Diameter (cm)	Inhibition Ratio (%)
*Angelica dahurica* (Fisch. ex Hoffm.) Benth. et Hook. f. ex Franch. et Sav	*Apiaceae Lindl.*	root	(6.07 ± 0.12) ^d^	4.56
*Angelica pubescens* Maxim. f. biserrata Shan et Yuan	*Apiaceae Lindl.*	root	(3.42 ± 0.23) ^o^	38.93
*Angelica sinensis*	*Apiaceae Lindl.*	root	(2.98 ± 0.2) ^q^	51.31
*Atractylodes macrocephala* Koidz.	*Asteraceae Bercht. & J. Presl*	rhizome	(4.28 ± 0.37) ^n^	23.57
*Cinnamomum cassia* Presl	*Lauraceae Juss.*	bark	(2.02 ± 0.03) ^r^	66.99
*Cnidium monnieri* (L.) Cuss.	*Apiaceae Lindl.*	fruit	(4.58 ± 0.11) ^kl^	24.55
*Coptis chinensis* Franch.	*Ranunculaceae Juss.*	total plant	(3.3 ± 0.17) ^p^	41.07
*Crataegus pinnatifida* Bunge	*Rosaceae Juss.*	fruit	(4.35 ± 0.19) ^mn^	22.32
*Cynanchum otophyllum*	*Paeoniaceae Raf.*	root	(6.45 ± 0.02) ^c^	-
*Eucommia ulmoides* Oliver	*Eucommiaceae Engl.*	bark	(5.4 ± 0.02) ^g^	15.09
*Forsythia suspensa*	*Oleaceae*	fruit	(4.62 ± 0.26) ^k^	17.5
*Glycyrrhiza uralensis* Fisch.	*Fabaceae Lindl.*	rhizoma	(5.28 ± 0.015) ^h^	13.73
*Heartleaf Houttuynia* Herb	*Saururaceae Rich. ex T. Lestib.*	leaves	(6.58 ± 0.21) ^b^	-
*Isatis tinctoria*	*Brassicaceae Burnett*	rhizoma	(6.8 ± 0.16) ^a^	-
*Mentha haplocalyx* Briq.	*Labiatae*	leaves	(4.42 ± 0.02) ^m^	21.07
*Morus alba* L.	*Moraceae Gaudich.*	shoot	(5.02 ± 0.17) ^j^	10.36
*Reynoutria japonica* Houtt.	*Polygonaceae*	rhizome	(5.62 ± 0.05) ^ef^	11.64
*Rheum palmatum* L.	*Polygonaceae*	rhizoma	(4.5 ± 0.29) ^l^	29.25
*Rhizoma Pinelliae*	*Araceae Juss.*	stem tuber	(5.22 ± 0.11) ^h^	6.79
*Rhus chinensis* Mill.	*Anacardiaceae R. Br.*	Galls on leaves	(0) ^s^	100
*Salvia miltiorrhiza* Bunge	*Labiatae*	root	(5.07 ± 0.53) ^i^	20.28
*Saposhnikovia divaricata* (Trucz.) Schischk.	*Apiaceae Lindl.*	root	(5.63 ± 0.17) ^e^	7.24
*Syzygium aromaticum* (L.) Merr. EtPerry	*Myrtaceae Juss.*	fruit	(0) ^s^	100
*Xanthium sibiricum Patrin ex Widder*	*Asteraceae Bercht. & J. Presl*	fruit	(5.54 ± 0.05) ^f^	8.73

Notes: Data in the table are mean value ± standard deviation. The different letters in the same column indicate significant differences at the 0.05 levels. Positive control: Jinggangmycin EC_50_ = 206.76 μg/mL; Negative control: Sterile water.

**Table 2 molecules-26-01667-t002:** Indoor toxicity of ethanol extracts from two plants to *R. solani*.

Concentration (µg/mL)	The Ethanol Extracts from Clove		Concentration (µg/mL)	The Ethanol Extracts from Gallnut	
	Colony Diameter (cm)	Inhibition Ratio (%)		Colony Diameter (cm)	Inhibition Ratio (%)
0	(6.85 ± 0.1) ^a^	-	0	(6.85 ± 0.1) ^a^	-
10	(5.62 ± 0.2) ^b^	17.96	15	(6.56 ± 0.07) ^b^	4.23
20	(3.85 ± 0.2) ^c^	43.8	30	(3.75 ± 0.1) ^c^	45.26
30	(2.69 ± 0.12) ^d^	60.73	45	(3.22 ± 0.11) ^d^	52.99
40	(1.38 ± 0.11) ^e^	79.85	60	(2.7 ± 0.05) ^e^	60.29
50	(0.68 ± 0.15) ^f^	89.49	75	(1.57 ± 0.03) ^f^	77.66

Notes: Data in the table are mean value ± standard deviation. The different letters in the same column indicate significant differences at the 0.05 level.

**Table 3 molecules-26-01667-t003:** Toxic regression equations of ethanol extracts from two plants to *R. solani*.

Plants	Toxic Regression Equation (Y = A + B × X)	Correlation Coefficient	EC_50_ (µg/mL)	EC_75_ (µg/mL)
gallnut	Y = −0.6717 + 3.5590x	0.9678	41.84	72.59
clove	Y = 0.0250 + 3.9031x	0.9593	21.68	36.34

**Table 4 molecules-26-01667-t004:** Biocontrol efficiency of ethanol extracts from gallnut and clove against rice sheath blight pathogen *R. solani* under green house condition.

Ethanol Extract from Gallnut			Ethanol Extract from Clove		
Treatment	Disease Index	Disease Suppression (%)	Treatment	Disease Index	Disease Suppression (%)
0	3.59		0	3.59	
200 µg/mL	2.92	18.66 ^e^	200 µg/mL	3.26	9.19 ^e^
400 µg/mL	2.75	23.40 ^d^	400 µg/mL	2.96	17.55 ^d^
600 µg/mL	2.47	31.20 ^c^	600 µg/mL	2.83	21.17 ^c^
800 µg/mL	2.32	35.38 ^b^	800 µg/mL	2.56	28.69 ^b^
1000 µg/mL	1.91	46.80 ^a^	1000 µg/mL	2.23	37.88 ^a^

Notes: Data in the table are mean value ± standard deviation. The different letters in the same column indicate significant differences at the 0.05 levels.

**Table 5 molecules-26-01667-t005:** MRM ion pairs and mass spectral parameters for eugenol and lauric acid.

Compounds	ESI	Parent (m/z)	Daughter (m/z)	Cone voltage/V	Colision energy/eV
eugenol	-	163	148	29	13
			121		24
lauric acid	-	259	245	75	30
			199		45

## Data Availability

Not applicable.

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
