# Peer review of "Growth Restriction of Rhizoctonia solani via Breakage of Intracellular Organelles Using Crude Extracts of Gallnut and Clove"

_molecules, 2021, doi:10.3390/molecules26061667_

Round 1

Reviewer 1 Report

I commend the authors for their interesting work. The paper is well-written and the experiment is excellent in my opinion. 

However, I have some minor comments:

- In the complete manuscript the scientific name of R. solani need to be italic and all scientific names as well. 

  • In the material and methods: The identification of the Rhizoctonia solani strain, need to be further explained, and the person or group of researchers who identified.
  • In Table 5: the cultivars of plant species cannot be italic, for example, in Cinnamomum cassia Presl , "Presl " cannot be italic. Please check that in all tables and figures.
  • In the conclusion, please show the novelty of the work as well as in the introduction. 
  • The conflict of interest section need to corrected. 

Author Response

Point 1:In the complete manuscript the scientific name of R. solani need to be italic and all scientific names as well.

Response 1: Firstly,I want to say “Thank you for  your comment”for sincerely,Since the manuscript was not carefully corrected, thank you very much for your suggestion. I have revised it according to your suggestion,the modified part marked by red font.

Point 2: In the material and methods: The identification of the Rhizoctonia solani strain, need to be further explained, and the person or group of researchers who identified.

Response 2: The strain was isolated and preserved by the Crop Protection Institute of Guizhou University. All preserved strains were sequenced and compared, the similarity between the results and Rhizoctonia solani AG1 IA was 100%. Thank you very much for your suggestion,the modified part marked by red font.

 Point 3:In Table 5: the cultivars of plant species cannot be italic, for example, in Cinnamomum cassia Presl , "Presl " cannot be italic. Please check that in all tables and figures.

Response 3: Thank you for your careful review.I carefully look up the names of plants and use the correct fonts to represent plant species,the modified part marked by red font.You are a very careful reviewer, and my article must be more perfect with your help.

 Point 4:In the conclusion, please show the novelty of the work as well as in the introduction.

Response 4:The conclusion part is refined again, which completes the creation of highlights. Thank you for your suggestion.I  have marked the specific content in red font in the conclusion of the article,I hope to get more  suggestions  from you again.

“conclusion:Plant-sourced fungicides are a class of plant antibacterial substances or compounds that induce plant defenses. They can kill or inhibit the growth of target pathogens efficiently with no or low toxicity, degrade readily, do not prompt development of resistance, which has led to their widespread use. In this study, the growth inhibition effect of 24 plant-sourced ethanol extracts on hyphae of rice sheath blight was studied. Ethanol extract of gallnuts and cloves inhibited the growth of rice sprites by up to 100%. Indoor toxicity measurement results showed that the gallnut and glove constituents inhibition reached 39.23 μg/mL and 18.82 μg/mL, respectively.The results showed that protection of Rhizoctonia solani viability reached 46.8% for gallnut and 37.88% for clove in water emulsions of 1,000 μg/mL gallnut and clove in the presence of 0.1% Tween 80. The protection by gallnut was significantly stronger than that of clove on rice sheath blight. Although these extracts inhibit the mycelium of rice sheath blight and improve the plant defense, they have not yet been developed into pesticides for application in rice agriculture as suitable substitutes for commercial chemical fungicides. To achieve this, optimized field trials are needed in terms of effective sterilization to determine their application doses in paddy fields.”

Point 5: The conflict of interest section need to corrected.

Response 5:We correct conflict of interest part again, and all the project fund providers and all the authors have no objection,Thank you for your wisdom reminder.

Lastly,Thank you very much for your valuable advice. I will certainly learn a lot from it, which will be of great help to my future writing.You  are a great reviewer,Good  for you.

Reviewer 2 Report

The paper describes the effect of ethanol extracts from 24 plants on R. solani.  From this first assay, two extracts were selected for a deeper investigation about the morphological effects on fungus.  The obtained results showed extract from cloves with higher potential to use as fungicide.

Some corrections and improvements listed below are necessary on the paper.

Line 64: Authors described about chemical composition of plants: "only 10% have been studied concerning their chemical composition".  However, the reference used to estimate this percentual is from 1977.   Please, try to up to date this reference.

Line 68:  Please up to date the number of secondary metabolites described in the literature.  Use, for example, the data from Dictionary of Natural Products.

Lines 86 & 89 & 91: Microorganism and plant names should be written in italics.

Line 89:  Revise the phrase to plural. Two extracts were described.

Line 94: (a) The table title must be revised because much more than two plants were investigated.  (b) Additionally, which is the criteria to organize the table ? Please, organize the table by alphabetical order of plant names.  (c) Authors could verify the feasibility of inserting the botanical family names. (d) Plant identifier must be inserted in all cases OR removed from all names. Please check !  (e) What was the positive control used in the assay?  In table footnote, authors must indicated this data.

Line 153: 72.59 ug/mL

Line 247: Table 5 should be merged with table 1 generating an unique table.

Discussion of results was based on the presence of eugenol, a typical component from volatile oil of cloves. Probably this component is found too in ethanol extract. However, chromatographic procedures were not performed to ensure the presence of eugenol in the extract, as well as to define the relative percentage of this metabolite in the extract. Therefore, I strongly recommend the inclusion of this data on the paper.

Author Response

Point 1:Line 64: Authors described about chemical composition of plants: "only 10% have been studied concerning their chemical composition".  However, the reference used to estimate this percentual is from 1977.   Please, try to up to date this reference.

Response 1: Firstly,I want to say “Thank you for  your comment”for sincerely,I have changed the latest references and read the Dictionary of Natural  through the website,the modified part marked by red font.

Point 2: Line 68:  Please up to date the number of secondary metabolites described in the literature.  Use, for example, the data from Dictionary of Natural Products.

Response 2: I changed article like this “More than 500,000 plant species have been identified globally. However, Only about 10 % of known plants have been studied chemical composition[13]. Grange et al. reported that 2,400 plants have active ingredients for pest control [14]. Plant resources are extraordinarily rich in China, with more than 10,000 kinds of Chinese herbal medicine resources, 1,400 kinds of plants with inhibitory activity, and over 328,000 natural compounds[13,15].”the modified part use in red font. I really hope you give us more valuable opinions,I sincerely hope to get more help from you.

“13 Ji, Y.R.; Yang, Q.L.; Dong, Y.; Ma, Z.J.; Liu, S.X.; Zhang, Z.H.; Guan, X.J. Research and Application on nematicidal activity of the Leguminosae plants. Journal of Anhui Agri. Sci., 2014, 42:8884-8886, 8889. doi:10.13989/j.cnki.0517-6611.2014.26.006

15 Buckingham, J., Ed. Dictionary of Natural Products on DVD, Version 29.2. Chapman and Hall/CRC: Boca Raton, FL, USA, 2019.

  Point 3:Lines 86 & 89 & 91: Microorganism and plant names should be written in italics.

Response 3: Thank you for your careful review.I carefully look up the names of plants and use the correct fonts to represent plant species,the modified part marked by red font.You are a very careful reviewer, and my article must be more perfect with your help.

 Point 4:Line 89:  Revise the phrase to plural. Two extracts were described.

Response 4:Thank you for your suggestion.I  will present the phrase in the plural,,the modified part marked by red font,Thank you for carefully revising my article.

Point 5:Line 94: (a) The table title must be revised because much more than two plants were investigated.  (b) Additionally, which is the criteria to organize the table ? Please, organize the table by alphabetical order of plant names.  (c) Authors could verify the feasibility of inserting the botanical family names. (d) Plant identifier must be inserted in all cases OR removed from all names. Please check !  (e) What was the positive control used in the assay?  In table footnote, authors must indicated this data.

Response 5:You are a great  reviewer,(a)Because there is no serious correction of the article, there is make a mistake. Thank you for your correction. I will definitely take it seriously.(b)At  the beginning of writing ,I only want to record the order of data to arrange the names of plants, and  I forgot the standard of sorting by English letters. Thank you very much for your excellent suggestions.(c) In order to ensure the correctness of plant names,I reconfirmed the correct names of the plant family.(d)I use Latin instead of Plant identifier , which is convenient for reference and unique. The Latin name adopts Linnaeus's "double name method". You can know which genus it is and what the name is. Subspecies, varieties and variants will also be written. var. is a variety. Finally, there will be a naming person. Generally speaking, Latin names are the complete scientific names of plants. English names have no such specificity,It can also be that my understanding of your suggestion is not very accurate, so I hope to get more help from you.(e)Jinggangmycin is usually used as positive control in China. I have added this part of data according to your suggestion.

Point 6: Line 153: 72.59 ug/mL

Response 6:This value is the EC75 value of galla chinensis,It is the concentration that causes 75% hypha death.I chose EC75 as the concentration of  experiment is when I observed with microscope, Mycelia began to change abnormally under  this concentration ,so I finally chose EC75 instead of EC50 for scanning electron microscope and transmission electron microscope experiment.

Point 7:Line 247: Table 5 should be merged with table 1 generating an unique table.

Response 7: I have merged the two tables into one according to your suggestion, which is more concise and avoids duplication. This is really a good suggestion,I am very honored to let you help me marked  my article and let me learn very practical knowledge.

Point 8:Discussion of results was based on the presence of eugenol, a typical component from volatile oil of cloves. Probably this component is found too in ethanol extract. However, chromatographic procedures were not performed to ensure the presence of eugenol in the extract, as well as to define the relative percentage of this metabolite in the extract. Therefore, I strongly recommend the inclusion of this data on the paper.

Response 8: I extracted eugenol from the volatile oil of my clove extract. I analyzed the content of eugenol by liquid chromatography-mass spectrometry, That demonstrated the hypothesis I talked about with this data, you make our  article  more exciting,Thank you for you help,You are an excellent artist, and article has become more interesting after your polishing.

“ LC-MS and data analysis:The quantitative ion chromatogram of lauric acid was obtained by liquid chromatography-mass spectrometry (Figure 5a). the peak time of lauric acid was about 7.584 min, the regression equation was Y=327272.248x-254692.2965, the correlation coefficient was 0.9987, which indicated that there was a good linear relationship between the concentration and the corresponding peak area in the concentration range of 10-100 μg /mL.The content of lauric acid in ethanol extract of Gallnut (10mg/mL) is 1.39 mg/mL, Therefore, the content of lauric acid in ethanol extract of Galla chinensis is about 13.9%.

The quantitative ion chromatogram (Figure 5b) of eugenol was obtained by liquid chromatography-mass spectrometry. The peak time of eugenol was 4.012 min. The regression equation was Y = 67959x+40242, the correlation coefficient was 0.9988, which indicated that there was a good linear relationship between the concentration and the corresponding peak area in the range of 1-100 μg /mL. The concentration of clove extract was 10mg/mL, Therefore,the detection of eugenol was 4.43 mg/mL,the content of eugenol in clove extract is about 44.3%.

Lastly,Thank you very much for your valuable advice. I will certainly learn a lot from it, which will be of great help to my future writing.You  are a great reviewer,Good  for you.

Round 2

Reviewer 2 Report

Authors made the changes suggested by the reviewers.

Figure 5 quality must be improved.